# Chemical Probes and Activity-Based Protein Profiling for Cancer Research

**DOI:** 10.3390/ijms23115936

**Published:** 2022-05-25

**Authors:** Mohammad Faysal Al Mazid, Seung Bin Park, Subba Rao Cheekatla, Dhiraj P. Murale, Kyung Ho Shin, Jun-Seok Lee

**Affiliations:** 1Division of Bio-Medical Science and Technology, Korea Institute of Science and Technology School, Korea University of Science and Technology, Seoul 02792, Korea; faysal@chembiol.re.kr; 2Department of Pharmacology, College of Medicine, Korea University, Seoul 02792, Korea; 2018010562@korea.ac.kr (S.B.P.); subbarao@chembiol.re.kr (S.R.C.); kyungho@korea.ac.kr (K.H.S.); 3Center for Advanced Biomolecular Recognition, Korea Institute of Science and Technology, Seoul 02792, Korea; dhiraj.murale@chembiol.re.kr

**Keywords:** activity-based probe, near-infrared probe, photodynamic therapy

## Abstract

Chemical probes can be used to understand the complex biological nature of diseases. Due to the diversity of cancer types and dynamic regulatory pathways involved in the disease, there is a need to identify signaling pathways and associated proteins or enzymes that are traceable or detectable in tests for cancer diagnosis and treatment. Currently, fluorogenic chemical probes are widely used to detect cancer-associated proteins and their binding partners. These probes are also applicable in photodynamic therapy to determine drug efficacy and monitor regulating factors. In this review, we discuss the synthesis of chemical probes for different cancer types from 2016 to the present time and their application in monitoring the activity of transferases, hydrolases, deacetylases, oxidoreductases, and immune cells. Moreover, we elaborate on their potential roles in photodynamic therapy.

## 1. Chemical Probes and Activity-Based Profiling

Chemical probes are “small molecules to understand the function of genes and proteins and their roles in physiology and pathology” (Chem. Biol., 2010, 17, 561–577) by either perturbation or monitoring the function of gene or protein [1,2]. The analysis of biological functions remains a challenging sector for understanding the roles of different proteins in disease incidence and progression. Identification of disease-related probes can facilitate a better understanding of the pathophysiological mechanisms of diseases. In particular, probes that can be utilized as disease-specific markers can promote a more favorable prognosis [3]. The rapid development of chemical biology tools for understanding cancer genomics, regulation, stage monitoring, and metastasis regulation has facilitated the scientific investigation of oncogenic signaling pathways and recovery processes [4]. Activity-based protein profiling (ABPP) is a chemical proteomics approach that utilizes chemical probes to understand the functional activity state of enzymes from the intact biological environment [5,6]. Based on the intermediate state of enzymatic reaction, chemical probes that produce irreversible chemical bonds using electrophilic warhead motifs, depending on the enzyme activity, are dubbed activity-based probes (ABPs). ABPP is a fundamental approach for synthesizing potential probes based on their target proteins that facilitate the determination of their roles in cellular systems and their impact on disease-controlling mechanisms. Structurally, ABPs consist of a reactive group (which covalently binds to the active site of the protein of interest), a binding tag (which attaches the reactive group with a reporter tag), and a labelling group (to detect the reactive group). The reactive group of probes contains an electrophilic site that attaches to the nucleophilic active site of the protein of interest. The reporter tag facilitates the visualization, identification, and quantification of labelled proteins through microscopy, sodium dodecyl-sulfate polyacrylamide gel electrophoresis (SDS-PAGE) separation, and mass spectrometric analysis. Generally, ABP reporter tags consist of fluorophores to visualize labelled proteins during analysis and microscopy. Affinity tags are usually biotin-coated tags that facilitate in the analysis of labelled proteins by mass spectrometry or Western blotting (Figure 1) [7]. On the other hand, affinity-based probes (AfBPs) consist of ligands for target binding and photoreactive groups for covalent bond formation, instead of electrophile warheads. The conjugation of a reporter tag with the target protein is primarily conducted by bio-orthogonal ligation, for example, of alkynes and azides, through click chemistry (Figure 1) [8]. These probes must possess a highly light-responsive element or photoreactive group that demonstrates fluorogenic properties upon UV irradiation by forming a covalent bond with the protein of interest. Several types of chemical activations, such as those involving selenium (Se), disulfide (S–S), and thioester bonds, have been used to conduct photoaffinity-based probe proteomics. For instance, Se can form C–Se or Se–Se covalent bonds; however, C–Se bonds can be broken by H_2_O_2_. For example, Yang et al. synthesized a C–Se-based photo-crosslinker that can cleave and capture peptide pools from chaperones [9]. In the bio-orthogonal-based probe ligation process, a tag is attached to the probe to facilitate downstream functional analysis, and a handle is added to bind small molecules and their binding partner proteins (Figure 1). Click reactions can be applied to activity-based probes, enzyme inhibitor profiles, and hybrid monolithic columns for proteomic analysis and protein labeling. Both approaches are widely popular for detecting targets and monitoring their activities. Major click chemistry reactions involve copper (I)-catalyzed azide–alkyne cycloaddition (CuAAC), strain-promoted azide–alkyne cycloaddition (SPAAC), inverse-electron-demand Diels–Alder (IEDDA) reaction, and Staudinger ligation [10]. The identification of numerous probes will facilitate a better understanding of diseases, especially of cancer. In this review, we discuss the development of chemical probes for monitoring disease progression and cancer stage determination and elaborate on the applicability of photodynamic therapy (PDT) from 2016 onwards.

## 2. Transferase Activity Probe

Kinases belong to a subgroup of transferase enzymes that are capable of transferring phosphate groups to acceptor proteins. Exploring the functions of kinases is important for understanding the potential roles of cancer signaling pathways. Small cell lung cancer (SCLC) involves several kinases that play a role in cancer progression. Desthiobiotin-ATP probes were developed to target kinase regulators that occur in 21 SCLC cell lines. The results revealed that, in SCLC cells, aurora kinase B is a critical kinase for MYC amplification, and TANK-binding kinase 1 (TBK1) is crucial for cancer cell viability, G2/M arrest, and apoptosis [11]. Zhang et al. synthesized a red-emitting probe called NB-BF to monitor Pim-1 kinase trafficking in cancer cells in vivo [12]. Moreover, tyrosine kinase Mer (MERTK) is a potential drug target for cancer and plays a role in the enhancement of the metastasis signaling cascade in tumor cells. Interestingly, a near-infrared fluorescent (NIR) molecular probe for Mer (MERi-SiR) was developed to monitor metastatic conditions in a mouse model system (Figure 2B) [13]. G-protein-coupled receptors (GPCRs) are key regulatory factors in cancer prognosis. Contextually, a photoreactive probe, namely, LEI121, was used to target type 2 cannabinoid receptor (CB2R), a class of GPCR; LEI121 covalently binds to CB2R. Upon photoactivation of this probe, it binds to endogenously expressed CB2R in HL-60 cells to monitor CB2R activity (Figure 2A) [14]. Bush et al. developed a series of photoaffinity probes (P1-P10) for selective binding to members of the cyclin-dependent kinase (CDK) family [15]. A series of clickable probes, namely HX03, HX04, and HX05, were synthesized to bind to epithelial growth factor receptor–tyrosine kinase inhibitors (EGFR-TKIs) found in non-small cell lung cancer (Figure 2C) [16]. To detect HER1/HER2 expression, Cy3-AFTN and Cy5-AFTN dual-targeting probes were developed to facilitate the diagnostic imaging of cancer cells in vivo (Figure 3) [17]. Similarly, to enable targeting of the B cell receptor (BCR) signaling mediator Bruton’s tyrosine kinase, a dual-purpose probe, IB-4, with both imaging and inhibitory activities, was developed [18].

Glutathione transferase (GST) is a well-known metabolic enzyme that is highly abundant in cancer cells. GST regulates several metabolic pathways within the cell and triggers drug resistance. Therefore, monitoring GST levels is important for cancer regulation and therapy. Cui et al. utilized boron–dipyrromethene (BODIPY) dye, a two-photon probe BNPA, to monitor GST activities in both cancer cells [19]. Similarly, Changjun et al. also developed a fluorescent probe, Cy-GST, to detect GST concentrations in cells and mouse models. They applied the Cy-GST probe to monitor GST levels in pulmonary fibrosis cells in mouse models, and the probe showed strong selectivity in both in vitro and in vivo systems [20]. In addition, Feng et al. reported a two-photon probe, P-GST, to detect GST levels in the setting of drug-induced liver injury [21]. Subsequently, Li et al. developed a luminescent probe (NRh-NNBA) to target GST in acute liver injury [22]. More recently, Liu et al. proposed a bifunctional fluorescent probe, resorufin (RP), for GST activity monitoring and visualization. They used resorufin as a fluorophore, which can target GST as a substrate and provide enhanced fluorescence, in the probe (Figure 4B) [23]. Another class of probes, namely, NRh-NDs, was also synthesized to bind to GST with selectivity and sensitivity in U87, MCF-7, and A549 cell lines [24].

## 3. Hydrolase Activity Probe

After cardiac surgery, serine hydrolase enzymes can promote acute kidney injury in patients. An activity-based probe, fluorophosphonate-TAMRA, was synthesized to monitor serine hydrolase activity after surgery. By utilizing this probe, kallikrein-1 was found to be one of the key factors responsible for acute kidney injury [25]. Ubiquitin carboxy-terminal hydrolase L1 (UCHL1) is one of the key elements that promotes cancer progression as well as neurodegenerative diseases [26]. A class of cyanimide-based probes, 8RK64, with higher affinity and selectivity to bind with UCHL1 over others, such as UCHL3/5, has been introduced. Activity-based protein profiling by MS confirmed a better interaction of the probe with UGHL1 compared with that with other mutants (from cys to ala). This BODIPY-labeled probe can selectively target UCHL1 and emit fluorescence in zebrafish embryos and cancer cells (Figure 5) [26]. Another class of orthogonal probes, IMP-1710, also exhibited selective targeting of UCHL1. This class of probes is labeled with a catalytic cysteine residue, with tenfold-enhanced fluorescence selectivity over the 8RK64 probe. Chemical proteomic analysis showed FGFR2 to be an off-target example of this minimal extent of selectivity. This probe activity showed the promising activity on blocking profibrotic response in idiopathic pulmonary fibrosis [27]. Another 18F-labeled, activity-based probe ([18F]JW199) was developed to target the cancer-associated serine hydrolase NCEH1, particularly in triple-negative breast cancer [28].

Proteases are a class of hydrolase enzymes that play a role in protein breakdown in the cellular system. Therefore, developing specific protease probes is also potentially important for disease studies, including those on cancer. Garrison et al. suggested that cysteine protease inhibitors of neurotensin receptor subtype 1 (NTSR1) improve tumor progression in NTSR1-positive cancers [29]. A series of NIR-activity-based probes (NIRF-ABPs) have been developed to facilitate cysteine monitoring inside the cellular microenvironment. These probes used Cy5 as a dye with a very low background emission signal for imaging. One of the quenched probes, GB137, was synthesized to detect cysteine protease activity in vivo. The probe showed strong selectivity for cysteine proteases in both clinical and in vitro profiling [30]. Recently, Lee et al. developed ABPs to detect high-temperature requirement A (HTRA), a serine protease. Misfolded HTRA proteins are abundantly observed in Alzheimer’s disease (AD), Parkinson’s disease (PD), apoptotic signaling, and cancer cell invasion. Application of this probe to mitochondria has shown significant efficacy in monitoring HTRA2 in living cells [31]. Verhelst et al. recently developed an activity-based probe utilizing alkyne-substituted benzoxazin-4-ones for serine protease activity monitoring [32]. γ-glutamyl transpeptidase (GGT) is a threonine-type protease and a well-known marker of cancer cell progression. An NIR fluorescent probe, in which glutathione (GSH) was used as a recognition unit and NIR hemicyanine as a fluorophore, was developed to monitor GGT in living cells [33]. An NIR-GGT fluorescent probe, Cy-GSH, was developed to visualize the location of cancer cells in vivo, and it has potential uses in fluorescence-guided cancer surgery (Figure 6) [34]. A fluorescent turn-on probe (QI-PG-Glu) was developed to monitor GGT activities, particularly in lung cancer (A549 cells). This probe emits the HQI fluorophore, which intervenes in rRNA biogenesis by triggering the p53 signaling pathway. This regulation inhibits RNA polymerase I transcription activity to induce cancer cell apoptosis [35]. A GGT-activatable fluorescent probe was synthesized using a covalently linked BODIPY fluorophore to monitor the GGT levels in different tumor cells. This probe successfully detected GGT levels in HeLa cells and a mouse model system [36]. Another GGT probe, called acetylation of chloro-rhodamine (ClR-Ac), shows sensitive fluorescence properties for peptidase in GGT bioimaging of living tumor cells [37]. CD13/aminopeptidase N (APN) is a key factor that promotes tumor growth, migration, and metastasis. A two-photon NIR fluorescence probe, DCM-APN, was developed to investigate APN function in cancer cells in vitro and in vivo [38]. Leucine aminopeptidase (LAP) is a proteolytic enzyme known as a biomarker for cancer and other diseases. A LAP-targeting NIR fluorescent probe, DCM-Leu, was developed with an NIR-emitting fluorophore (DCM) as a reporter tag. This probe triggers the L-leucine moiety and can be used to investigate LAP activity in different types of cells [39]. Later, a new class of probes containing an NIR-fluorophore, CHMC-M-Leu, was developed to monitor endogenous LAP activity in living cells [40]. Prolyl aminopeptidase (PAP) is an exopeptidase that can be used as a biomarker in pathogenic infections and several cancers. An NIR turn-on fluorescent probe, NIR-PAP, was developed by combining a cysteine-proline dipeptide with an acryloylated NIR fluorophore to identify and target PAP activity in living cells [41].

## 4. Oxidoreductase Probe

Pan et al. recently developed a fluorogenic probe, DNS-pE2, to monitor endogenous 3-phosphoglycerate dehydrogenase (PHGDH) activity in mammalian cells. PHGDH is an enzyme that plays a role in serine biosynthesis and enhances serine concentration in cancer cells. MCF-7 breast cancer cells pretreated with the probe were used to validate the utility of the probe for PHGDH activity monitoring [42]. Nitric oxide (NO) is a well-known tumor prognostic biomarker that plays a role in modulating the reactive nitrogen oxide species (RNS) content of the tumor microenvironment, leading to metastasis [43]. A genetically encoded SNAP-tag probe, TMR-NO-BG, was developed to label fused proteins with high NO sensitivity in the mitochondrial inner membrane, nucleus, and cytoplasm [44]. Calcium (Ca^2+^) influx is an important factor in the anti-apoptotic pathway that suppresses ROS generation in the cancer microenvironment [45]. A fluorescent probe, SA-4CO_2_Na, was developed to detect Ca^2+^ ions in the millimolar range (0.6–3.0 mM) with a significant distinguishing capacity between hypercalcemic (1.4–3.0 mM) and normal (1.0–1.4 mM) Ca^2+^ ion levels in cancer cells [46]. A probe, NF-O-SBD, was synthesized to detect cysteine function in ROS-induced cells. This probe showed a very low detection limit for Cys/Hcy under cellular oxidative conditions [47]. Another mitochondrial ROS generation probe, HQPQ-B, was developed to target H_2_O_2_ influx [48]. Zhang et al. synthesized a dual-activatable NIR-II molecular probe, PN910, to monitor H_2_O_2_ and ONOO^−^ in alkaline tissues [49]. A new class of far-red fluorescent probes, Mito-TG, increased mitochondrial matrix ROS generation levels in different live cells [50]. More recently, Peng et al. developed a BODIPY-based fluorescence probe, BODIPY-T, for sensing O_2_ radical dot-level changes in RAW264.7 cells [51]. The combination of the aldoxime reaction group with rhodamine fluorophore led to the development of an NIR fluorescent probe, specifically HClO. This probe showed higher sensitivity towards ROS generation to distinguish between cancer cells and normal tissues [52].

## 5. Epigenetics Regulator Activity Probe

High levels of class I histone deacetylases (HDACs) are essential epigenetic regulators associated with cancer cell proliferation and metastasis. An NIR imaging probe, 10-Cy5.5, was developed to monitor and visualize epigenetic changes mediated by class I HDACs, particularly in TNBC MDA-MB-231 cells and xenograft tumor models. The probe was subjected to suberoylanilide hydroxamic acid (SAHA), which subsequently downregulated probe uptake by tumor cells. This result indicates the therapeutic efficacy of the probe in monitoring HDAC activity [53]. Tian et al. developed an NIR probe called IRDye800CW-labeled SAHA (IRDye800CW-SAHA) to detect HDAC levels in hepatocellular carcinoma. The probe showed promising efficacy in detecting overexpression of HDACs in hepatocellular carcinoma [54]. A bioluminescence “turn-on” probe, AcAH-Luc (6-acetamidohexanoic acid–d-luciferin), was also developed for monitoring esterase (CES) and HDAC activity in malignant tumors and normal cells. The application of the AcAH-Luc probe in vitro showed that the limits of detection (LODs) of the probe were 0.495 nM for esterase and 1.14 nM for HDAC [55]. Zhang et al. also developed an NIR fluorescent probe, CyAc-RGD, for imaging HDAC6 in cancer cells and under in vivo conditions. This probe was applied to HeLa cells to selectively detect HDAC6. The in vivo mouse model showed a downregulation of probe uptake in the presence of SAHA (Figure 7B). This experiment validated the utilization of the probe in targeted drug-response monitoring [56].

## 6. Anticancer Activity of Probes

Identification of bioactive molecules is a fundamental approach in the development of therapeutic strategies. Affinity-based protein profiling has facilitated the identification of bioactive molecules in disease studies. Ma et al. reported two photoreactive anticancer inhibitors, arenobufagin and HM30181. Upon UV irradiation, a covalent cross-link is formed with the protein of interest. These probes identify PARP1 as the hit compound using ABPP and cellular imaging processes [57]. A quantitative proteomic approach revealed that DR is an inhibitor of discoidin domain receptor 1 (DDR1). A series of trans-cyclooctene (TCO)-containing probes can be used to monitor the binding interactions of DR inhibitors with their targets within the cell. This proteomics approach revealed that cathepsin D (CTSD) is the main off-target of DR in human cancer cells [58]. In parallel, OSW-1 is a potent anticancer saponin that has a positive effect on breast cancer cells [59]. Moreover, OSW-1 is known to activate PI3K-AKT signaling in hepatocellular carcinoma cells (e.g., SK-Hep1) [60]. A photoactive diaziridine group was incorporated into OSW-1 to produce an effective anticancer agent with photophysical properties [61]. A noteworthy case involves the utilization of bioactive compounds as AfBPs for proteomic studies. Indeed, Lang et al. successfully developed two probes for the detection of entinostat and camptothecin and used them for SILAC-based target identification and cellular imaging [62].

## 7. Immune Cell Activity Probe

Human neutrophil elastase (HNE) promotes immune inflammation, ultimately leading to lung cancer. An NIR fluorescent probe, F-1, has been developed to achieve HNE monitoring in cells. The probe showed promising activities for monitoring HNE activity in lung cancer cells (A549 cells) [63]. Similarly, an activity-based non-peptide ratiometric fluorescent probe, DCDF(I), was developed to monitor HNE in A549 and HeLa cells. The results were verified using HPLC and HRMS spectra, where this probe showed the lowest detection limit compared with that of the other available HNE probes [64]. Chang et al. recently synthesized a NeutropG probe for the specific identification and imaging of active neutrophils in fresh blood samples (Figure 8) [65]. BODIPY chromophores were used to develop a fluorogenic probe that could form J-aggregates and emit turn-on signals in the presence of eosinophil peroxidase. This probe can be utilized to detect oxidative stress in tumor cells and mouse immune responses against eosinophil peroxidase [66]. Metastatic lymph nodes (MLNs) are a sign of the early stages of tumor metastasis. A matrix metalloproteinase-2 (MMP-2)-activatable probe was constructed using Cy5 dye to monitor and visualize tumor-induced lymphangiogenesis [67]. Through the combination of two probes, venetoclax (targeting B cell lymphoma or Bcl-2 [Probe-ABT-199]) and idasanutlin (targeting MDM2 [RG7388]), a new probe was synthesized with promising anticancer efficacy in vitro and in vivo. Moreover, dual imaging of Bcl-2/MDM2 facilitated the detection of apoptosis marker proteins ITPR1, GSR, RER1, PDIA3, Apoa1, and Tnfrsf17 by LC–MS/MS in various cancer cells [68]. Another class of MDM2-targeted probes, RG7388 (idasanutlin), was developed using an ^18^F-labeled radiotracer to determine MDM2 expression in tumors [69].

## 8. Chemical Probe for Photodynamic Therapy

Bio-orthogonal reactions are widely used in photodynamic therapy to optimize phototoxicity as well as specific sub-cellular localization. Halogenated BODIPY-tetrazine (mTz-2I-BODIPY) probes were synthesized to obtain specific PDT therapy in cellular nuclei, which leads to singlet O_2_ generation, resulting in cancer cell death (Figure 9) [70]. A heterogeneous copper catalyst probe utilizing CuAAC processes can activate fluorophores in biological systems, such as cells and zebrafish [71]. Moreover, to overcome adverse drug reactions, in situ drug delivery systems have been developed by combining enzyme-instructed supramolecular self-assembly (EISA) and (Tz)–trans-cyclooctene (Tz/TCO) using the bio-orthogonal decaging reaction. The authors suggested that a combination of enzymatic and bio-orthogonal reactions can resolve adverse reactions during chemotherapy [72]. The development of highly effective anticancer drugs that cause minimal damage to the surrounding normal tissues remains a challenge in cancer therapy. A photothermal agent, i.e., the dual-targeted organic molecule Bio-PPH3-PT, plays a role in targeting the mitochondria of malignant cells, promoting the death of these cells [73]. A lipid nanoparticle-embedded small-molecule NIR fluorophore and a quencher attached to a PtdEtn moiety, pyropheophorbide *a*-phosphatidylethanolamine-QSY21 (Pyro-PtdEtn-QSY), have been described to target choline metabolism of phosphatidylcholine-specific phospholipase C (PC–PLC) in breast, prostate, and ovarian cancers [74]. Another photothermal agent (MAL-CDYQ) can destroy bioactive proteins in cells through heat generation [75]. Similarly, a fluorescence-guided photodynamic therapy (FL-PDT), involving a lipid-droplet-targeted TPECNPB, has been developed for H_2_O_2_-activatable fluorescence-guided PDT of cancer cells [76]. Moreover, a pH-dependent amino heptamethine cyanine-based theragnostic probe (I_2_-IR783-Mpip) can be activated by near-infrared light in acidic conditions by generating singlet oxygen [77]. Yuan et al. developed a multifunctional luminescent probe to monitor hydrogen sulfide (H_2_S) during PDT therapy in cancer cells [78]. Another NIR chemiluminescent probe, CL-SO, was synthesized to selectively monitor singlet oxygen in cells and in a mouse model during PDT therapy [79].

## 9. Current Challenges and Future Perspective

Chemical probe tools offer highly reliable analyses of the biological complexity of cancer through monitoring and understanding the key regulatory factors with a dynamic range of fluorogenic properties. Another broad application of probe proteomics is in the identification of cancer biomarkers. In this approach, AfBP facilitates target identification by phenotypic screening rather than by observing and monitoring activities. For example, in the cellular system, ANXA2, PDIA3/4, FLAD1, and NOS2 targets show hindering properties. A series of tetrazole-based probes have been used to identify these targets by phenotypic screening and affinity-based proteome profiling [80]. In this review, we discussed the applicability of chemical probes in cancer biology factor imaging over the past five years. A series of new kinases, hydrolases, proteases, deacetylases, transferases, immune cells, and ROS/RNS generation activity monitoring probes have been identified in the last few years with improved photo-physical properties. The newly identified kinase-targeted probes for Myc-binding, TANK-binding kinase 1 (TBK1), Mer (MERi-SiR), GPCR, and CDKs are potentially beneficial for a better understanding of the kinase regulators in cancer. A series of newly developed UCHL1, GGT, GST, and HDAC probes have also displayed improved applicability in both in vitro and cellular systems. Monitoring of the key immune system players using probes helps researchers in facilitating immunotherapy and enhancing post-immunotherapy outcomes in numerous ways. The recent advances in probes allow better monitoring of the cancerous environment; however, the analyses of biocompatibility, cytotoxicity, and immunogenic responses remain fundamental challenges to be overcome. In general, small molecules requires to meet several criteria to act as a chemical probe, including being a potent modulator or inhibitor of certain biochemical function, where in vitro activity of probe should be less than 100 nM with half-maximal inhibitory concentration IC_50_; selectivity should be 30-fold higher than other competitor portion and should have reasonable cell penetrant with proper on target selectivity [81]. In addition, several online platforms are enlarging to identify and determine the probe characteristics along with their applications. Some of them are PubChem (https://pubchem.ncbi.nlm.nih.gov, accessed on 4 May 2022), ChEMBL (https://www.ebi.ac.uk/chembl/, accessed on 4 May 2022), Probe miner (http://probeminer.icr.ac.uk/#/, accessed on 4 May 2022), SGC chemical probe collection (https://www.thesgc.org/chemical-probes, accessed on 4 May 2022), Guide to Pharmacology (https://www.bps.ac.uk/publishing/guide-to-pharmacology, accessed on 4 May 2022), and Chemical Probe Portal (http://www.chemicalprobes.org/, accessed on 4 May 2022). Chemical probes have improved the current PDT approaches for studying tumor incidence and progression as well as the therapeutic inventions in cancer. These novel probes aid researchers in understanding the disease-causing mechanisms in more sophisticated and precise ways; however, considerably better probe selection needs to be a focus for further development in this field. To analyze hypertoxicity, a series of new probes with better physical properties and reduced levels of immunotoxicity are needed. It is expected that improvements in the overall applicability of probes will contribute to the scientific community’s pursuit of better clinical treatment options and therapies against cancer.

## Figures and Tables

**Figure 1 ijms-23-05936-f001:**
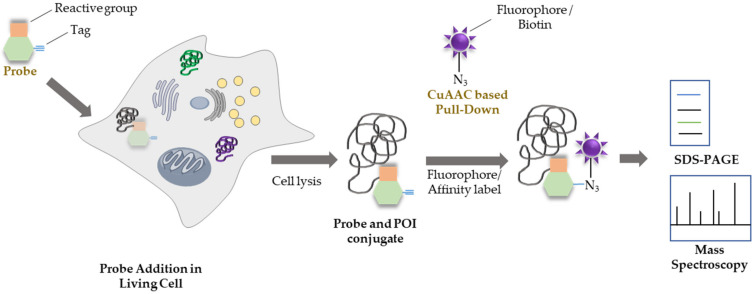
**Mode of action of a chemical probe in cancer proteomics**. CuAAC—copper (I)-catalyzed azide–alkyne cycloaddition; POI—protein of interest; SDS-PAGE, sodium dodecyl-sulfate polyacrylamide gel electrophoresis.

**Figure 2 ijms-23-05936-f002:**
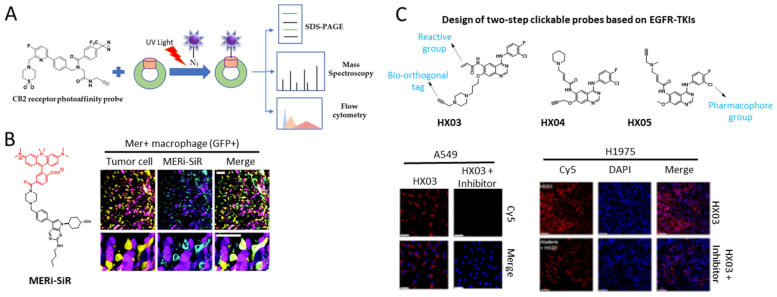
**Probe for Kinase monitoring.** (**A**) Schematic diagram of chemical probe binding interaction with CB2 receptor and analysis procedures; (**B**) structure of MERi-SiR probe and confocal imaging of tissue in metastatic cancer. (**B**) Reprinted with permission from [9]. Copyright 2018, The Royal Society of Chemistry. (**C**) Chemical structure of HX03, HX04, and HX05 probes (top panel), fluorescence imaging of HX03 probe activity on A549 cell and tumor tissue of xenograft mice models. (**C**) Reprinted with permission from [12]. Copyright 2022, Elsevier.

**Figure 3 ijms-23-05936-f003:**
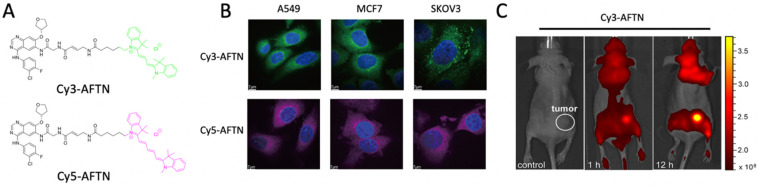
**Cy3/5 AFTN probe for HER1/2 Kinase monitoring.** (**A**) Chemical structure of Cy3/5 AFTN probe. (**B**) Confocal images of Cy3/5 AFTN probe in A549, MCF-7, and SKOV3 cells to monitor HER1/2 activity. C. Fluorescence images of Cy3-AFTN-treated xenograft mice. (**B**,**C**) Reprinted with permission from (ref. [13]). Copyright 2018, American Chemical Society.

**Figure 4 ijms-23-05936-f004:**
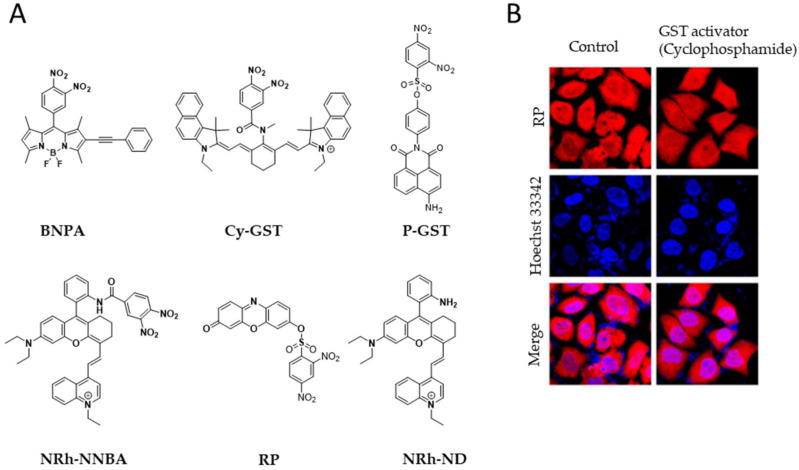
**Glutathione transferase (GST) detecting probes.** (**A**) Chemical structure of glutathione transferase (GST)-detecting probes. (**B**) Confocal images of RP probe activity for GST monitoring in HepG2 cells. (**B**) Reprinted with permission from [19]. Copyright 2020, Elsevier.

**Figure 5 ijms-23-05936-f005:**
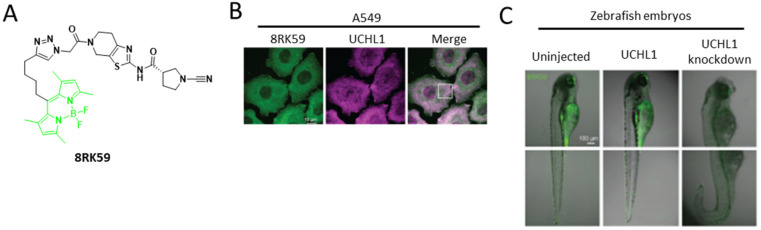
**Hydrogenase activity monitoring.** (**A**) Structure of 8RK59 probe. (**B**) Immunofluorescence staining of UCHL1 in an 8RK59-labeled control and shUCHL1 A549 cells. (**C**) Monitoring UCHL1 activity in zebrafish embryos with/without 8RK59 probe. (**B**,**C**) Reprinted with permission from [22]. Copyright 2020, American Chemical Society.

**Figure 6 ijms-23-05936-f006:**
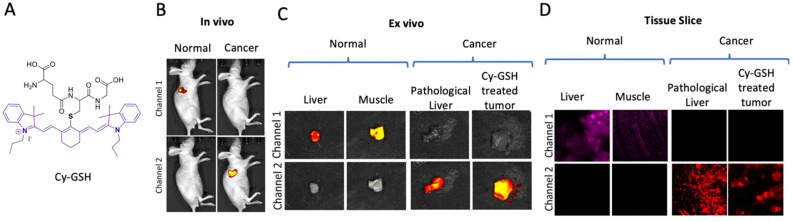
**Cy-GSH probe for threonine-type protease, GGT targeted image guiding surgery.** (**A**) Chemical structure of Cy-GSH probe. B-D. In vivo, Ex vivo and Tissue slice of Cy-GSH probe activity for GGT monitoring. (**B**–**D**) Reprinted with permission from [30]. Copyright 2018, American Chemical Society.

**Figure 7 ijms-23-05936-f007:**
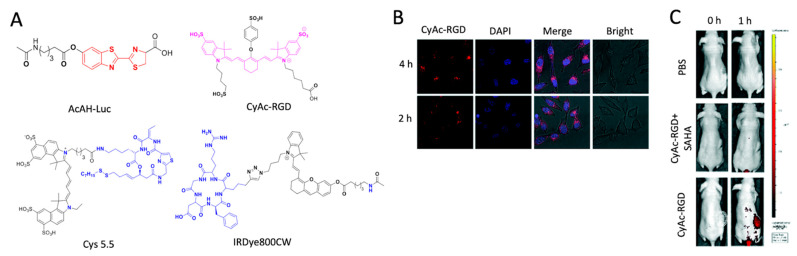
**Histone deacetylases (HDACs) detecting probes.** (**A**) Chemical structures of histone deacetylases (HDACs)-detecting probes. (**B**) Fluorescence images of HeLa cells incubated with CyAc-RGD probe. (**C**) Imaging of CyAc-RGD probe in tumor induced mice. (**B**,**C**) Reprinted with permission from [52]. Copyright 2018, The Royal Society of Chemistry.

**Figure 8 ijms-23-05936-f008:**
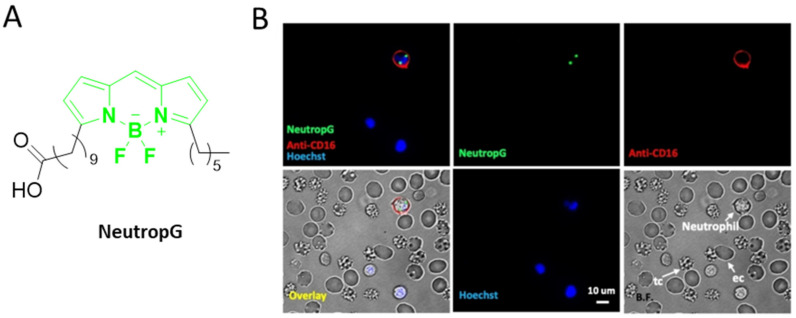
Neutrophil-detecting probes. (**A**) Structure of NeutropG probe. (**B**) NeutropG probe selectivity in whole blood without RBC lysis. (**B**) Reprinted with permission from [61]. Copyright 2021 Wiley-VCH GmbH.

**Figure 9 ijms-23-05936-f009:**
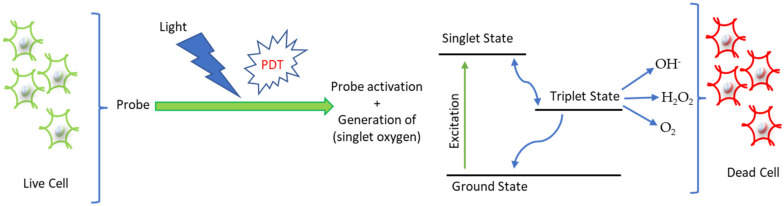
Application of chemical probes in photodynamic therapy (PDT). The photosensitive probe absorbs light and changes the ground-state energy status to single-state energy. By intersystem crossing energy exchange, it reaches the excited triplet state and generates OH^−^, H_2_O_2_, and O_2_, which ultimately lead to cancer cell death.

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
