# Peer review of "Chemical Probes and Activity-Based Protein Profiling for Cancer Research"

_ijms, 2022, doi:10.3390/ijms23115936_

Round 1
Reviewer 1 Report
The manuscript entitled “Chemical Probes and Activity-based Protein Profiling for Can- 2 cer Research” is indeed a great idea for a review article in the field of chemical biology. The authors have summarized many chemical probes identified for targeting various critical proteins involved in many crucial diseases. I hope my suggestions help improve the content. The whole manuscript should be revised for proper English.
Many sentences are obscure and generalized, I suggest the authors add more descriptive sentences in order to give depth in presenting each probe,for example:
“The identification of activity-based probes (ABPs) is a fundamental ap- 31 proach for synthesizing potential probes based on their target proteins”
“which helps in the analysis”: grammer
“different kinases in cancer biological systems” very general sentences.
Figure2, structure of all probes mentioned in the manuscript should be shown clearly and should be legible
(Bottom Left Panel), and many other examples: inconsistency in using capital and small letters,
Providing references for as many sentences as possible, for example for : “Ubiquitin carboxy- 135 terminal hydrolase L1 (UCHL1) is one of the key elements that promotes cancer progres- 136 sion as well as neurodegenerative diseases.”
“his probe activity was validated by breast cancer cell activity profiling”: more explanation needed
“Therefore, developing specific protease probes is crucial for disease 155 studies, including those on cancer.”: very general statements. Authors should avoid writing sentences as part of report. The text should come together as a whole. All these proteins are criticlal biological targets.
Authors should present a graph (not table) to summarize all probes, including their characteristics, that have seen designed for different proteins.
Choice of colour for chemical structures of probes is not consistent nor wise throughout the manuscript. Please consider editing the coulors/structures of probes.
Author Response
Reviewer 1
The manuscript entitled “Chemical Probes and Activity-based Protein Profiling for Cancer Research” is indeed a great idea for a review article in the field of chemical biology. The authors have summarized many chemical probes identified for targeting various critical proteins involved in many crucial diseases. I hope my suggestions help improve the content. The whole manuscript should be revised for proper English.
Many sentences are obscure and generalized, I suggest the authors add more descriptive sentences in order to give depth in presenting each probe, for example:
R1-Q1. “The identification of activity-based probes (ABPs) is a fundamental approach for synthesizing potential probes based on their target proteins”
R1-A1: Thank you very much for valuable suggestion. As per the request from the reviewer, we restructured the statement considering the fate of those proteins (line 33 to 38)
R1-Q2. “which helps in the analysis”: grammar
R1-A2: As the reviewer suggested, we have rephrased the statement (line 47)
R1-Q3. “different kinases in cancer biological systems” very general sentences.
R1-A3: We have removed this statement from the context.
R1-Q4. Figure2, structure of all probes mentioned in the manuscript should be shown clearly and should be legible (Bottom Left Panel), and many other examples: inconsistency in using capital and small letters,
R1-A4: Thank you very much for improving the flow, as per the suggestion we have changed the figure labels and legends.
R1-Q5. Providing references for as many sentences as possible, for example for: “Ubiquitin carboxyterminal hydrolase L1 (UCHL1) is one of the key elements that promotes cancer progression as well as neurodegenerative diseases.”
R1-A5: We have included the reference for the statement (line 142).
R1-Q6. “his probe activity was validated by breast cancer cell activity profiling”: more explanation needed
R1-A6: We have restated this segment with better application of this probe (line 152).
R1-Q7. “Therefore, developing specific protease probes is crucial for disease 155 studies, including those on cancer.”: very general statements. Authors should avoid writing sentences as part of report. The text should come together as a whole. All these proteins are critical biological targets.
R1-A7: We have rephrased the statement (line 162)
R1-Q8. Authors should present a graph (not table) to summarize all probes, including their characteristics, that have seen designed for different proteins.
R1-A8: We have uploaded the graphical figure to summarize the whole concept of this manuscript
R1-Q9. Choice of color for chemical structures of probes is not consistent nor wise throughout the manuscript. Please consider editing the colors/structures of probes.
R1-A9: Thank you very much for your suggestion. We changed the color of the figures as per your comment.

Reviewer 2 Report
The paper is updated and concise study.
The English should be revised by English language expert/professional.
Author Response
Reviewer 2
The paper is updated and concise study.
R2-Q1.The English should be revised by English language expert/professional.
R2-A1: Thank you very much for your comment. We have included the certificate for English editing for your consideration.

Reviewer 3 Report
The review discusses the synthesis of chemical probes for different cancer types. It involves publications from 2016 to the present time. The review speaks about the use of the probes for monitoring the activity of transferases, hydrolases, deacetylases, and oxidoreductases. Potential roles of probes in photo-dynamic therapy are also reported. The review includes 76 references.
Although the review targets “chemical probes” and “activity-based protein profiling”, these terms are not defined in the manuscript. Neither is disclosed the history of these terms and are cited earlier important review publications on these topics.
Actually, “chemical probes” are small organic compounds with special biological properties.
Specialists (e.g., Victoria Vu, et al., https://doi.org/10.1146/annurev-biochem-032620-105344) have outlined criteria that a small molecule chemical probe should meet. First, the compound should be a potent modulator (usually an inhibitor) of the biochemical function of the target protein, displaying an in vitro activity [half maximal inhibitory concentration (IC50)] of less than 100 nM. Second, the compound should be at least 30-fold selective for the target over other sequence-related proteins within the same target family and not be promiscuous or broadly reactive. Finally, a chemical probe should be reasonably cell penetrant, as evidenced by significant on-target cellular activity at 1 μM.
Unfortunately, several of the reported in the present review manuscript fluorescent compounds are ligands of enzymes or receptors whose affinity towards non-target proteins has not been tested or it has been tested towards a very limited number of proteins. Therefore, they cannot be considered as “chemical probes” or specific ABPP tools. This is misleading for the audience of the review. The chosen examples do not correspond to strict terms associated with “chemical probes” and possess limited value for biomedical studies or diagnostic tests. The authors have not performed critical analysis of published data and use the interpretations provided by authors of the studies
The quality of certain bioactive compound, chemical probes, can be controlled in several open-access databases (e.g., https://www.ChemicalProbes.org).
The topic of probes in photo-dynamic therapy is not directly related to other topics and it should not be a part of the review. Also, there are some inaccuracies introduced into the discussion of this topic [which leads to O2 generation, resulting in cancer cell death].
Not explained: what are photogenic and fluorogenic probes?
Author Response
Reviewer 3
The review discusses the synthesis of chemical probes for different cancer types. It involves publications from 2016 to the present time. The review speaks about the use of the probes for monitoring the activity of transferases, hydrolases, deacetylases, and oxidoreductases. Potential roles of probes in photo-dynamic therapy are also reported. The review includes 76 references.
R3-Q1. Although the review targets “chemical probes” and “activity-based protein profiling”, these terms are not defined in the manuscript. Neither is disclosed the history of these terms and are cited earlier important review publications on these topics.
R3-A1: We appreciate the point by the reviewer. In this revised manuscript, we added the definition of the terms and cited relevant review article particularly to clarify the terms (chemical probe: ref 1,2, activity-based protein profiling: ref 5,6).
R3-Q2. Actually, “chemical probes” are small organic compounds with special biological properties. Specialists (e.g., Victoria Vu, et al., https://doi.org/10.1146/annurev-biochem-032620-105344) have outlined criteria that a small molecule chemical probe should meet. First, the compound should be a potent modulator (usually an inhibitor) of the biochemical function of the target protein, displaying an in vitro activity [half maximal inhibitory concentration (IC50)] of less than 100 nM. Second, the compound should be at least 30-fold selective for the target over other sequence-related proteins within the same target family and not be promiscuous or broadly reactive. Finally, a chemical probe should be reasonably cell penetrant, as evidenced by significant on-target cellular activity at 1 μM. Unfortunately, several of the reported in the present review manuscript fluorescent compounds are ligands of enzymes or receptors whose affinity towards non-target proteins has not been tested or it has been tested towards a very limited number of proteins. Therefore, they cannot be considered as “chemical probes” or specific ABPP tools. This is misleading for the audience of the review.
The chosen examples do not correspond to strict terms associated with “chemical probes” and possess limited value for biomedical studies or diagnostic tests. The authors have not performed critical analysis of published data and use the interpretations provided by authors of the studies. The quality of certain bioactive compound, chemical probes, can be controlled in several open-access databases (e.g., https://www.ChemicalProbes.org).
R3-A2: Thank you very much for your insightful comments. As the reviewer pointed out, there are broad ranges of chemical probes from highly specific ones to promiscuous ones. Because both kind of probes have their unique utilities, we tried to collected compounds related to cancer related application. In fact, some of case studies are the early stage of chemical probe developments for cancer study, but we believe these will be worth to mentioned in the review for a milestone for next research. By vigorous scientific progression, probe characteristics as well their applications can be improved. Many more emerging probes newly appeared in the field of cancer procurement arena. Therefore, we tried to merge those potential probes to lead the readers for further progression in cancer procurement.
However, it is still important to mention about the criteria of ideal chemical probe as the reviewer mentioned for robust and practical usage. Thus, we carefully added a paragraph to cover the information. In addition, we also mentioned about the several open-access databases to get spec of chemical probes (line 355 to 365). In this way, we believe readers will be able to balance and compare chemical probes. We really appreciate the reviewer’s valuable comments.
R3-Q3. The topic of probes in photo-dynamic therapy is not directly related to other topics and it should not be a part of the review. Also, there are some inaccuracies introduced into the discussion of this topic [which leads to O2 generation, resulting in cancer cell death].
R3-A3: Thanks for the reviewer’s comment. In the field of cancer biology, we believe PDT is notably emerging field to appeal more and more chemical biologist. Therefore, we believe it would be great to introduce the PDT therapy and anticancer agents related chemical probe/sensors, and worth to be covered in the review.
Regarding the inaccurate statements, we fixed the contents to clarify the meaning in line 306. We really appreciate the reviewer’s help to clarify the meaning.
R3-Q4. Not explained: what are photogenic and fluorogenic probes?
R3-A4: We appreciate the reviewer’s point. To avoid ambiguity of the meaning, we removed all terms “photogenic” as this terminology is not fully established term. With this revision, we believe overall readability significantly improved in this revised manuscript. Thank you very much for valuable points from the reviewer.

Round 2
Reviewer 2 Report
The revised paper is ok for publication.
Author Response
Thank you very much for the reviewer's positive consideration.
Reviewer 3 Report
The authors have added definitions of chemical probes now. Also, references to relevant publications have been added. Thereafter, I hope that the authors acknowledge that they are not reviewing chemical probes. These compounds are mostly rather “more or less selective small-molecule tracers used for fluorescent imaging”.
Please explain what is the difference of probes notified as AfBP, ABP, and ABPP.
Wording of definitions should be improved.
Some figures are corrupted in my file (Fig. 2, Fig.4).
Author Response
Reviewer 3
The authors have added definitions of chemical probes now. Also, references to relevant publications have been added. Thereafter, I hope that the authors acknowledge that they are not reviewing chemical probes. These compounds are mostly rather “more or less selective small-molecule tracers used for fluorescent imaging”.
R3-Q1. Please explain what is the difference of probes notified as AfBP, ABP, and ABPP.
R3-A1. We appreciate the reviewer’s valuable comment. We now added the definitions included in line number 35 to 63 to clarify the meanings. After this revision, overall readability improves a lot. We truly appreciate the comment.
R3-Q2. Wording of definitions should be improved.
R3-A2. Thank you very much for your comment. We further revised the sentences to clarify the meaning and get professional English Editing Service.
R3-Q3. Some figures are corrupted in my file (Fig. 2, Fig.4).
R3-A3. After checking the file, we re-uploaded the figure files for your consideration.